# Implicit Contrastive Representation Learning with Guided Stop-gradient

**Byeongchan Lee**[*]
Gauss Labs
Seoul, Korea
byeongchan.lee@gausslabs.ai

**Sehyun Lee**[*]
KAIST
Daejeon, Korea
sehyun.lee@kaist.ac.kr

## Abstract

In self-supervised representation learning, Siamese networks are a natural architecture for learning transformation-invariance by bringing representations of positive pairs closer together. But it is prone to collapse into a degenerate solution. To address the issue, in contrastive learning, a contrastive loss is used to prevent collapse by moving representations of negative pairs away from each other. But it is known that algorithms with negative sampling are not robust to a reduction in the number of negative samples. So, on the other hand, there are algorithms that do not use negative pairs. Many positive-only algorithms adopt asymmetric network architecture consisting of source and target encoders as a key factor in coping with collapse. By exploiting the asymmetric architecture, we introduce a methodology to implicitly incorporate the idea of contrastive learning. As its implementation, we present a novel method guided stop-gradient. We apply our method to benchmark algorithms SimSiam and BYOL and show that our method stabilizes training and boosts performance. We also show that the algorithms with our method work well with small batch sizes and do not collapse even when there is no predictor. The code is available in the supplementary material.

## 1 Introduction

Representation learning has been a critical topic in machine learning. In visual representation learning, image representations containing high-level semantic information (e.g., visual concept) are learned for efficient training in downstream tasks. Because human annotation is labor-intensive and imperfect, learning representations without labels is getting more attention. In many cases, unsupervised or self-supervised learning (SSL) has surpassed its supervised counterpart.

In SSL [Jaiswal et al., 2020, Jing and Tian, 2020, Liu et al., 2021], there are roughly two branches of algorithms. One branch is a set of algorithms trained on pretext tasks with pseudo labels. Examples of the pretext tasks are predicting relative positions [Doersch et al., 2015], solving jigsaw puzzles [Noroozi and Favaro, 2016], colorization [Zhang et al., 2016], and identifying different rotations [Gidaris et al., 2018]. However, relying on a specific pretext task can restrict the generality of the learned representations. The other branch is a set of algorithms trained by maximizing the agreement of representations of randomly augmented views from the same image. Many algorithms in this branch adopt Siamese networks with two parallel encoders as their architecture. Siamese networks are natural architecture with a minimal inductive bias to learn transformation-invariant representations. However, naïve use of Siamese networks can result in collapse, i.e., a constant representation for all images.

To tackle the collapse problem, there have been four strategies. The first strategy is contrastive learning [Bachman et al., 2019, Ye et al., 2019, Chen et al., 2020a,b, He et al., 2020, Misra and Maaten, 2020, Chen et al., 2021]. Contrastive learning uses positive pairs (views of the same

---

[*]These authors contributed equally to this work.

37th Conference on Neural Information Processing Systems (NeurIPS 2023).

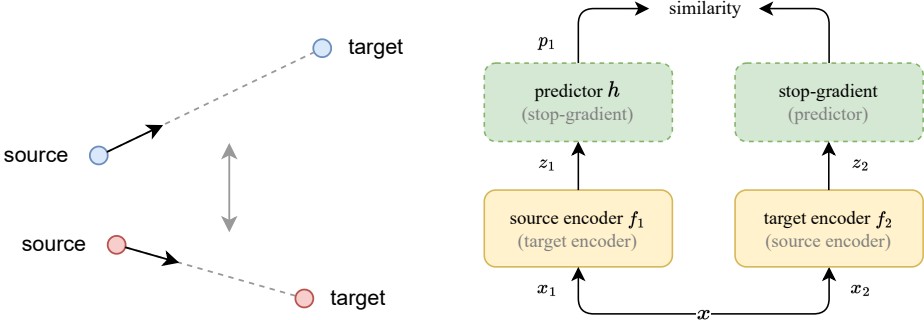

(a) Implicit contrastive representation learning.  (b) Symmetrized use of asymmetric networks.

Figure 1: (a) Dots of the same color are representations of a positive pair. Without contrastive loss, it aims for a repelling effect by carefully determining which to make the source representation and which to make the target representation. (b) In SimSiam and BYOL, a given image $x$ is randomly transformed into two views $x_1$ and $x_2$. The views are processed by encoders $f_1$ and $f_2$ to have projections $z_1$ and $z_2$. A predictor is applied on one side, and stop-gradient is applied on the other. Then, the similarity between the outputs from both sides is maximized. By using the predictor and stop-gradient alternately, a symmetric loss is constructed.

image) and negative pairs (views from different images). Minimizing contrastive loss encourages representations of negative pairs to push each other while representations of positive pairs pull each other. The second strategy is to use clustering [Caron et al., 2018, Asano et al., 2019, Caron et al., 2019, 2020, Li et al., 2020]. It clusters the representations and then predicts the cluster assignment. Repeating this process keeps the cluster assignments consistent for positive pairs. The third strategy is to decorrelate between other variables of representations while maintaining the variance of each variable so that it does not get small [Bardes et al., 2021, Zbontar et al., 2021]. The fourth strategy is to break the symmetry of Siamese networks [Caron et al., 2020, Grill et al., 2020, He et al., 2020, Caron et al., 2021, Chen and He, 2021]. Many algorithms using this strategy make the representation from one encoder (called source encoder) follow the representation from the other encoder (called target encoder)[2].

In this paper, we introduce a methodology to do contrastive learning implicitly by leveraging asymmetric relation between the source and target encoders. For a given positive pair, one of its representations becomes the source representation (from the source encoder), and the other becomes the target representation (from the target encoder). The target representation attracts the source representation. By investigating how representations are located in the embedding space, we carefully determine which representation will be the source (or target) representation so that representations of negative pairs repel each other (Figure 1a). There is no explicit contrastive part in our loss, thus the name implicit contrastive representation learning. The main idea of the methodology can be expressed as follows: *Repel in the service of attracting*. We also present our guided stop-gradient method, an instance of the methodology. Our method can be applied to existing algorithms SimSiam [Chen and He, 2021] and BYOL [Grill et al., 2020]. We show that by applying our method, the performance of the original algorithms is boosted on various tasks and datasets.

The technical contributions of our work can be summarized as follows:

- We introduce a new methodology called implicit contrastive representation learning that exploits the asymmetry of network architecture for contrastive learning (Section 3 and 4).

- We present new algorithms by applying our method to benchmark algorithms SimSiam and BYOL and show performance improvements in various tasks and datasets (Section 5).

- We demonstrate through empirical studies that our idea can be used to improve training stability to help prevent collapse, which is a fundamental problem of SSL (Section 6).

---

[2]The source encoder is also called online, query, or student encoder, and the target encoder is also called key or teacher encoder in the literature depending on the context [Wang et al., 2022, Tarvainen and Valpola, 2017, Grill et al., 2020, He et al., 2020, Caron et al., 2021].

## 2 Related work

**Siamese networks** Siamese networks are symmetric in many respects. There are encoders of the same structure on both sides, and they share weights. The inputs to the two encoders have the same distribution, and the outputs are induced to be similar by the loss. On the one hand, this symmetric structure helps to learn transformation-invariance well, but on the other hand, all representations risk collapsing to a trivial solution. There have been several approaches to solving this collapse problem, and the approaches related to our work are contrastive learning and asymmetric learning. So, we introduce them in more detail below.

**Contrastive learning** Contrastive learning [Hadsell et al., 2006, Wang and Isola, 2020, Wu et al., 2018, Hjelm et al., 2018, Tian et al., 2020] can be characterized by its contrastive loss [Le-Khac et al., 2020, Chopra et al., 2005]. The basic idea is to design the loss so that representations of positive pairs pull together and representations of negative pairs push away. Through the loss, representations are formed where equilibrium is achieved between the pulling and pushing forces. Contrastive losses that have been used so far include noise contrastive estimation (NCE) loss [Gutmann and Hyvärinen, 2010], triplet loss [Schroff et al., 2015], lifted structured loss [Oh Song et al., 2016], multi-class $N$-pair loss [Sohn, 2016], InfoNCE loss [Oord et al., 2018], soft-nearest neighbors loss [Frosst et al., 2019], and normalized-temperature cross-entropy loss [Chen et al., 2020a]. All the losses mentioned above contain explicit terms in their formula that cause the negative pairs to repel each other. Under self-supervised learning scenarios, since we don't know the labels, a negative pair simply consists of views from different images. Then, in practice, there is a risk that the negative pair consists of views from images with the same label, and performance degradation occurs due to this sampling bias [Chuang et al., 2020]. In addition, contrastive learning algorithms are sensitive to changes in the number of negative samples, so performance deteriorates when the number of negative samples is small.

**Asymmetric learning** Another approach to avoid collapse is to break the symmetry of Siamese networks and introduce asymmetry [Wang et al., 2022]. Asymmetry can be imposed on many aspects, including data, network architectures, weights, loss, and training methods. In MoCo [He et al., 2020], the encoders do not simply share weights, and the weights of one encoder (key encoder) are a moving average of the weights of the other encoder (query encoder). This technique of slowly updating an encoder is called a momentum encoder. SwAV [Caron et al., 2020] and DINO [Caron et al., 2021] apply a multi-crop strategy when performing data augmentation to make the distribution of inputs into encoders different. Also, the loss applied to the outputs from the encoders is asymmetric. Furthermore, in DINO, stop-gradient is applied to the output of one encoder (teacher encoder) and not the other (student encoder). In SimSiam [Chen and He, 2021] and BYOL [Grill et al., 2020], an additional module called predictor is stacked on one encoder (source encoder), and stop-gradient is applied to the output from the other encoder (target encoder). Compared to SimSiam, in BYOL, the target encoder is a momentum encoder.

## 3 Main idea

In this section, we explain our motivation and intuition behind our method using an example.

**Asymmetric architecture** In SimSiam and BYOL, for a given image $x$, two views $x_1$ and $x_2$ are generated by random transformations. The views are fed to encoders $f_1$ and $f_2$ to yield projections $z_1$ and $z_2$. An encoder is a backbone plus a projector[3]. Then, a predictor $h$ is applied to one encoder (source encoder) to have a prediction, and stop-gradient[4] is applied to the other encoder (target encoder). The algorithms maximize the similarity between the resulting prediction and projection. The difference between SimSiam and BYOL is in the weights of the encoders. In SimSiam, the source and target encoder share the weights. On the other hand, in BYOL, the target encoder is a momentum encoder. That is, the weights of the target encoder are an exponential moving average of the weights of the source encoder. Due to the existence of the predictor and stop-gradient (also momentum encoder in the case of BYOL), the algorithms have asymmetric architecture (Figure 1b).

---

[3]Ultimately, we use representations from the backbone, but it is common practice to compose the loss with projections from the projector, i.e., the encoder [Chen et al., 2020a].

[4]Applying stop-gradient to $z$ means treating $z$ as a constant. In actual implementation, $z$ is detached from the computational graph to prevent the propagation of the gradient.

**Symmetric loss** Despite the asymmetry of the architecture, the losses in SimSiam and BYOL are symmetrized. After alternately applying a predictor and stop-gradient to the two encoders, the following loss is obtained by adding the resulting loss terms:

$$\mathcal{L} = \frac{1}{2}\mathcal{D}(p_1, \text{sg}(z_2)) + \frac{1}{2}\mathcal{D}(p_2, \text{sg}(z_1)), \tag{1}$$

where $\mathcal{D}(\cdot, \cdot)$ denotes negative cosine similarity, i.e., $\mathcal{D}(p, z) = -(p/\|p\|_2) \cdot (z/\|z\|_2)$, and $\text{sg}(\cdot)$ denotes the stop-gradient operator. The range of possible values for the loss is $[-1, 1]$. Minimizing the first term brings $p_1$ (closely related to $z_1$) closer to $z_2$, and minimizing the second term brings $p_2$ (closely related to $z_2$) closer to $z_1$. By minimizing the loss, we want to move $z_1$ in the direction of $z_2$ and $z_2$ in the direction of $z_1$.

**Guided stop-gradient** The main idea of our guided stop-gradient method is to design an asymmetric loss to leverage the asymmetry of the architecture. In other words, we select one of the two loss terms in Equation (1) to help with training. However, it is known that randomly constructing an asymmetric loss is not beneficial [Chen and He, 2021]. To do this systematically, we consider two different images $x_1$ and $x_2$ simultaneously and let each other be a reference. When trying to bring the representations of one image closer, we give a directional guide on which representation to apply stop-gradient to by utilizing the geometric relationship with the representations of the other reference image.

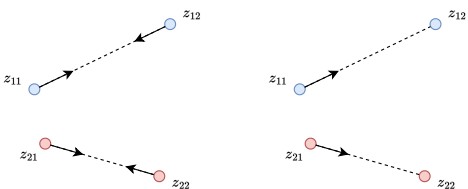

(a) Loss terms for two imgs.    (b) Selected loss terms.

Figure 2: An example for two images. The dots represent four projections of the two images. The arrows represent the expected effect of the loss terms. We want dots of the same color to come close to each other. We select loss terms so that two closest dots with different colors will fall apart.

Specifically, let $z_{11}$ ($p_{11}$) and $z_{12}$ ($p_{12}$) be projections (predictions) from the image $x_1$, and $z_{21}$ ($p_{21}$) and $z_{22}$ ($p_{22}$) be projections (predictions) from the image $x_2$. Then, the symmetrized loss for two images will be as follows (Figure 2a):

$$\mathcal{L} = \frac{1}{4}\mathcal{D}(p_{11}, \text{sg}(z_{12})) + \frac{1}{4}\mathcal{D}(p_{12}, \text{sg}(z_{11})) + \frac{1}{4}\mathcal{D}(p_{21}, \text{sg}(z_{22})) + \frac{1}{4}\mathcal{D}(p_{22}, \text{sg}(z_{21})). \tag{2}$$

Now, we select one term from the first two terms (terms to maximize agreement between $z_{11}$ and $z_{12}$) and another term from the remaining two terms (terms to maximize agreement between $z_{21}$ and $z_{22}$). One view of the image $x_1$ and one view of the image $x_2$ form a negative pair. So we want the projections of the image $x_1$ and the projections of the image $x_2$ not to be close to each other. In the example of Figure 2, since $z_{11}$ and $z_{21}$ are closest, we apply a predictor[5] to $z_{11}$ and $z_{21}$ and apply stop-gradient to $z_{12}$ and $z_{22}$ to try to separate the projections. Then, the resulting loss will be as follows (Figure 2b):

$$\mathcal{L} = \frac{1}{2}\mathcal{D}(p_{11}, \text{sg}(z_{12})) + \frac{1}{2}\mathcal{D}(p_{21}, \text{sg}(z_{22})). \tag{3}$$

Selecting loss terms is equivalent to determining which of the two projections of each image to apply stop-gradient. Since we do this in a guided way by observing how the projections are located, we call the method Guided Stop-Gradient (GSG). In this way, by continuously moving toward representations that are not close together, representations of negative pairs are induced to spread well in the long run.

**Implicit contrastive representation learning** In our loss, there is no explicit part where the projections of a negative pair repulse each other. However, the loss we designed implicitly does it. We aim for a contrastive effect by making good use of the fact that the source projections go after the target projections in an asymmetric network architecture. Therefore, SimSiam and BYOL with GSG can also be viewed as a mixture of contrastive learning and asymmetric learning.

---

[5]In the case of SimSiam and BYOL, the presence of a predictor can make the interpretation tricky because when we move two close points $z$ and $z'$, we move them indirectly by moving $p = h(z)$ and $p' = h(z')$ through the predictor $h$. We assume that the predictor $h$ has a good regularity as a function, that is, if $\|z - z'\|_2$ is small, $\|h(z) - h(z')\|_2$ is also small. So, by trying to separate $p$ and $p'$, we separate $z$ and $z'$. Note that in practice, the predictor $h$ is usually an MLP with a few layers.

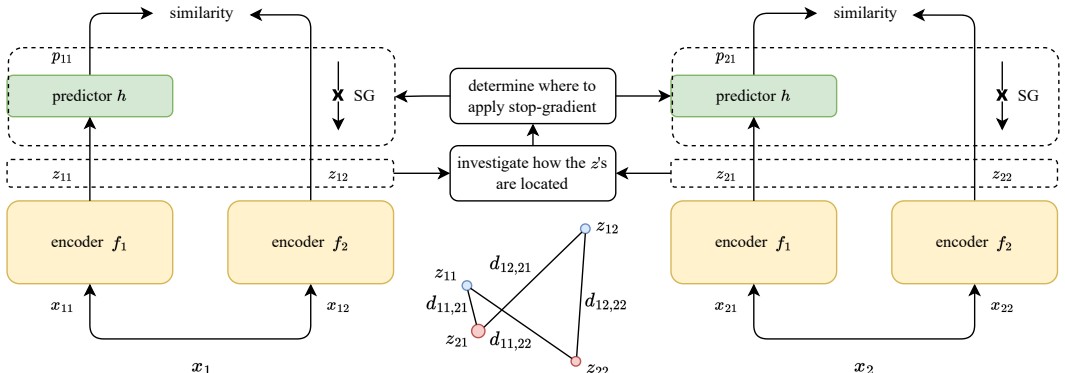

Figure 3: Overview of our guided stop-gradient method. (1) The encoders process two images $x_1$, $x_2$ that are reference to each other. (2) Investigate the distances $d_{11,21}$, $d_{11,22}$, $d_{12,21}$, and $d_{12,22}$ between the projections of negative pairs. (3) Determine which side to apply stop-gradient and which to apply a predictor.

## 4 Method

In this section, we explain the specific application of GSG based on the main idea described above.

For a given batch of images for training, a batch of pairs of images is created by matching the images of the batch with the images of the shuffled batch one by one. Then, a pair of images in the resulting batch consists of two images, $x_1$ and $x_2$. Note that since the images are paired within a given batch, the batch size is the same as the existing algorithm. By applying random augmentation, we generate views $x_{11}$ and $x_{12}$ from the image $x_1$ and views $x_{21}$ and $x_{22}$ from the image $x_2$. The views are processed by an encoder $f$ to yield projections $z_{11}$ and $z_{12}$ of the image $x_1$ and projections $z_{21}$ and $z_{22}$ of the image $x_2$.

Let $d_{ij,kl}$ denote the Euclidean distance between projections $z_{ij}$ and $z_{kl}$, i.e.,

$$d_{ij,kl} = \|z_{ij} - z_{kl}\|_2, \tag{4}$$

where $\|\cdot\|_2$ is $l_2$-norm. We investigate distances $d_{11,21}$, $d_{11,22}$, $d_{12,21}$, and $d_{12,22}$. Note that there are four cases in total since we are looking at the distance between one of the two projections of $x_1$ and one of the two projections of $x_2$. Now, we find the minimum $m$ of the distances. That is,

$$m = \min\{d_{11,21}, d_{11,22}, d_{12,21}, d_{12,22}\}. \tag{5}$$

We apply a predictor to the two projections corresponding to the smallest of the four distances and stop-gradient to the remaining two projections. Depending on which distance is the smallest, the loss is as follows:

$$\mathcal{L} = \begin{cases} \frac{1}{2}\mathcal{D}(p_{11}, \mathrm{sg}(z_{12})) + \frac{1}{2}\mathcal{D}(p_{21}, \mathrm{sg}(z_{22})), & \text{if } m = d_{11,21} \\ \frac{1}{2}\mathcal{D}(p_{11}, \mathrm{sg}(z_{12})) + \frac{1}{2}\mathcal{D}(p_{22}, \mathrm{sg}(z_{21})), & \text{if } m = d_{11,22} \\ \frac{1}{2}\mathcal{D}(p_{12}, \mathrm{sg}(z_{11})) + \frac{1}{2}\mathcal{D}(p_{21}, \mathrm{sg}(z_{22})), & \text{if } m = d_{12,21} \\ \frac{1}{2}\mathcal{D}(p_{12}, \mathrm{sg}(z_{11})) + \frac{1}{2}\mathcal{D}(p_{22}, \mathrm{sg}(z_{21})), & \text{if } m = d_{12,22}. \end{cases} \tag{6}$$

For a better understanding, refer to Figure 3 and Appendix A. For simplicity, we present the overview and pseudocode for SimSiam with GSG, but they are analogous to BYOL with GSG.

## 5 Comparison

In this section, we compare SimSiam and BYOL with GSG to the original SimSiam and BYOL on various datasets and tasks. For a fair comparison, we use the same experimental setup for all four algorithms on each dataset and task. For example, all algorithms perform the same number of gradient updates and exploit the same number of images in each update. We implement the algorithms with Pytorch [Paszke et al., 2019] and run all the experiments on 8 NVIDIA A100 GPUs. Our algorithms take about two days for ImageNet pre-training and 12 hours for linear evaluation.

Table 1: Comparison of representation quality under standard evaluation protocols.

| Algorithm | ImageNet | | CIFAR-10 | |
|---|---|---|---|---|
| | $k$-NN acc. (%) | Linear acc. (%) | $k$-NN acc. (%) | Linear acc. (%) |
| SimSiam | 51.7±0.11 | 67.9±0.09 | 77.0±0.67 | 82.7±0.26 |
| SimSiam w/ GSG | **58.4±0.17** | **69.4±0.02** | **82.2±0.48** | **86.4±0.28** |
| BYOL | 56.5±0.16 | 69.9±0.02 | 85.4±0.24 | 88.0±0.09 |
| BYOL w/ GSG | **62.2±0.06** | **71.1±0.12** | **89.4±0.21** | **90.3±0.16** |

Table 2: Comparison in transfer learning for image recognition.

| Algorithm | CIFAR-10 | Aircraft | Caltech | Cars | DTD | Flowers | Food | Pets | SUN397 | VOC2007 |
|---|---|---|---|---|---|---|---|---|---|---|
| SimSiam | 90.0 | 39.7 | 86.5 | 31.8 | 70.9 | 88.8 | 61.6 | 81.4 | 57.8 | 80.5 |
| SimSiam w/ GSG | **92.3** | **47.0** | **89.6** | **41.2** | **73.0** | **89.5** | **64.3** | **85.1** | **59.8** | **82.6** |
| BYOL | 91.0 | 42.5 | 88.9 | 39.3 | 71.7 | 89.1 | 64.0 | 85.3 | 60.4 | 82.2 |
| BYOL w/ GSG | **93.6** | **47.1** | **89.9** | **46.9** | **72.6** | **89.5** | **67.1** | **89.1** | **61.6** | **82.7** |

## 5.1 Pre-training

We first pre-train networks in an unsupervised manner. The trained networks will later be used in downstream tasks. We use ImageNet [Deng et al., 2009] and CIFAR-10 [Krizhevsky et al., 2009] as benchmark datasets. Refer to Appendix B for data augmentation details.

For ImageNet, we use the ResNet-50 backbone [He et al., 2016], a three-layered MLP projector, and a two-layered MLP predictor. We use a batch size of 512 and train the network for 100 epochs. We use the SGD optimizer with momentum of 0.9, learning rate of 0.1, and weight decay rate of 0.0001. We use a cosine decay schedule [Chen et al., 2020a, Loshchilov and Hutter, 2016] for the learning rate.

For CIFAR-10, we use a CIFAR variant of the ResNet-18 backbone, a two-layered MLP projector, and a two-layered MLP predictor. We use a batch size of 512 and train the network for 200 epochs. We use the SGD optimizer with momentum of 0.9, learning rate of 0.06, and weight decay rate of 0.0005. We do not use a learning rate schedule since no scaling shows better training stability. Refer to Appendix C for other implementation details.

After pre-training, we obtain representations of the images with the trained backbone and then evaluate their quality. We use both $k$-nearest neighbors [Wu et al., 2018] and linear evaluation, which are standard evaluation protocols.

$k$-**nearest neighbors** For a given test set image, we find its representation and obtain $k$ training set images with the closest representation. Then, we determine the predicted label of the image by majority voting of the labels of the $k$ images. We set $k = 200$ for ImageNet and $k = 1$ for CIFAR-10.

**Linear evaluation** We freeze the trained backbone, attach a linear classifier to the backbone, fit the classifier on the training set in a supervised manner for 90 epochs, and test the classifier on the test set. For ImageNet, we use a batch size of 4096 and the LARS optimizer [You et al., 2017], which can work well with a large batch size. For CIFAR-10, we use a batch size of 256 and the SGD optimizer with momentum of 0.9, learning rate of 30, and a cosine decay schedule. We report top-1 accuracy for all cases.

**Results** Table 1 shows that applying GSG consistently increases the performance. We report error bars (mean ± standard deviation) by running each algorithm three times independently and show that our method improves the performance reliably. The performance gain is obtained by keeping all other experimental setups the same and changing only the stop-gradient application method. This suggests that there is room for improvement in the symmetrized use of asymmetric networks in the existing algorithms.

Table 3: Comparison in transfer learning for object detection and semantic segmentation.

| Algorithm | VOC detection | | | COCO detection | | | Semantic segmentation | |
|---|---|---|---|---|---|---|---|---|
| | $AP_{50}$ | AP | $AP_{75}$ | $AP_{50}$ | AP | $AP_{75}$ | Mean IoU (%) | Pixel acc. (%) |
| SimSiam | 77.0 | 48.8 | 52.2 | 50.7 | 31.2 | 32.8 | 0.2626 | 65.63 |
| SimSiam w/ GSG | **79.8** | **51.2** | **55.1** | **53.2** | **33.4** | **35.6** | **0.3345** | **76.57** |
| BYOL | 79.2 | 50.3 | 54.5 | 52.0 | 32.5 | 34.5 | 0.2615 | 62.79 |
| BYOL w/ GSG | **80.5** | **52.0** | **56.4** | **53.7** | **33.8** | **36.1** | **0.2938** | **74.78** |

Table 4: Comparison between explicit and implicit contrastive learning algorithms.

| Methodology | Algorithm | $b = 1024$ | $b = 512$ | $b = 256$ |
|---|---|---|---|---|
| Explicit contrastive | End-to-end [He et al., 2020] | 57.3 | 56.3 | 54.9 |
| | InstDisc [Wu et al., 2018] | 54.1 | 52.0 | 50.0 |
| | MoCo [He et al., 2020] | 57.5 | 56.4 | 54.7 |
| | SimCLR [Chen et al., 2020a] | 62.8 | 60.7 | 57.5 |
| Implicit contrastive | SimSiam w/ GSG | 70.1 | 69.4 | 69.9 |
| | BYOL w/ GSG | 71.9 | 71.0 | 71.6 |

## 5.2 Transfer learning

An essential goal of representation learning is to obtain a pre-trained backbone that can be transferred to various downstream tasks. To evaluate whether our pre-trained backbones are transferable, we consider image recognition, object detection, and semantic segmentation tasks. We use the ResNet-50 backbones pre-trained on ImageNet. We follow the experimental setup in [Ericsson et al., 2021]. More implementation details can be found in Appendix D.

**Image recognition** We carry out image recognition tasks on different datasets. For datasets, we adopt widely used benchmark datasets in transfer learning such as CIFAR-10, Aircraft [Maji et al., 2013], Caltech [Fei-Fei et al., 2004], Cars [Krause et al., 2013], DTD [Cimpoi et al., 2014], Flowers [Nilsback and Zisserman, 2008], Food [Bossard et al., 2014], Pets [Parkhi et al., 2012], SUN397 [Xiao et al., 2010], and VOC2007 [Everingham et al., 2010]. These datasets vary in terms of the amount of data or the number of classes. We report the average precision AP at 11 recall levels $\{0, 0.1, ..., 1\}$ on VOC2007, mean per-class accuracy on Aircraft, Pets, Caltech, and Flowers, and top-1 accuracy on the rest of the datasets. For the evaluation protocol, we perform the linear evaluation.

**Object detection** We perform object detection tasks on Pascal-VOC [Everingham et al., 2010] and MS-COCO [Lin et al., 2014]. We use VOC2007 trainval as the training set and VOC2007 test as the test set. We report $AP_{50}$, AP, and $AP_{75}$. $AP_{50}$ and $AP_{75}$ are average precision with intersection over union (IoU) threshold 0.5 and 0.75, respectively. We freeze the pre-trained backbone except for the last residual block. We use a Feature Pyramid Network [Lin et al., 2017] to extract representations, and a Faster R-CNN [Ren et al., 2015] to predict. We do the experiments on the Detectron2 platform [Wu et al., 2019].

**Semantic segmentation** We conduct semantic segmentation tasks on MIT ADE20K [Zhou et al., 2019]. We use ResNet-50 as the encoder and use UPerNet [Xiao et al., 2018] (the implementation in the CSAIL semantic segmentation framework [Zhou et al., 2018, 2017]) as the decoder. It is based on Feature Pyramid Network and Pyramid Pooling Module [Zhao et al., 2017]. We train for 30 epochs and test on the validation set. We report mean IoU and pixel accuracy. For the predicted results of each algorithm, refer to Appendix F.

**Results** Table 2 and 3 show that applying GSG increases the performance consistently. We can conclude that the pre-trained backbones are transferable to different tasks, and our method helps to get better quality representations.

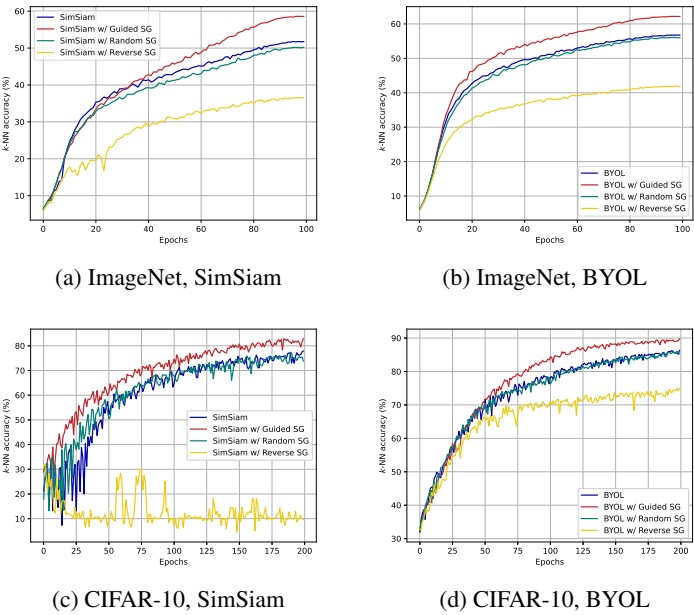

|  |  |
|:---:|:---:|
| (a) ImageNet, SimSiam | (b) ImageNet, BYOL |
| (c) CIFAR-10, SimSiam | (d) CIFAR-10, BYOL |

Figure 4: Importance of guiding. Depending on how stop-gradient is used, performance is significantly different. It shows the best performance when used along with GSG.

## 5.3 The number of negative samples

It is known that the performance of contrastive learning algorithms is vulnerable to reducing the number of negative samples [Tian et al., 2020, Wu et al., 2018, He et al., 2020, Chen et al., 2020a]. In this respect, we compare our algorithms to benchmark contrastive learning algorithms (all with the ResNet-50 backbone). Table 4 reports linear evaluation accuracy on ImageNet (the performance of the benchmark algorithms is from [He et al., 2020] and [Chen et al., 2020a]). End-to-end, SimCLR, and our algorithms use samples in the batch as negative samples, so for these algorithms, $b$ in the table denotes the batch size. InstDisc and MoCo maintain a separate memory bank or dictionary, so for these algorithms, $b$ in the table denotes the number of negative samples from the memory bank or dictionary. The table shows that our algorithms work well with small batch sizes.

## 6 Empirical study

In this section, we broaden our understanding of stop-gradient, predictor, and momentum encoder, which are components of SimSiam and BYOL by comparing variants of the algorithms.

## 6.1 Importance of guiding

We demonstrate that it is crucial to apply stop-gradient in a guided way by changing how stop-gradient is used. We compare our GSG with what we name random stop-gradient and reverse stop-gradient. In random stop-gradient, stop-gradient is randomly applied to one of the two encoders, and a predictor is applied to the other. That is, we randomly select one of the four equations in Equation (6). On the other hand, in reverse stop-gradient, stop-gradient is applied opposite to the intuition in GSG. In other words, we select the remaining loss terms other than the selected loss terms when GSG is used. In Equation (2), two loss terms will be selected to form the loss as follows:

$$\mathcal{L} = \frac{1}{2}\mathcal{D}(p_{12}, \text{sg}(z_{11})) + \frac{1}{2}\mathcal{D}(p_{22}, \text{sg}(z_{21})). \tag{7}$$

Therefore, the random stop-gradient is a baseline where stop-gradient is naïvely applied, and the reverse stop-gradient is the worst-case scenario according to our intuition.

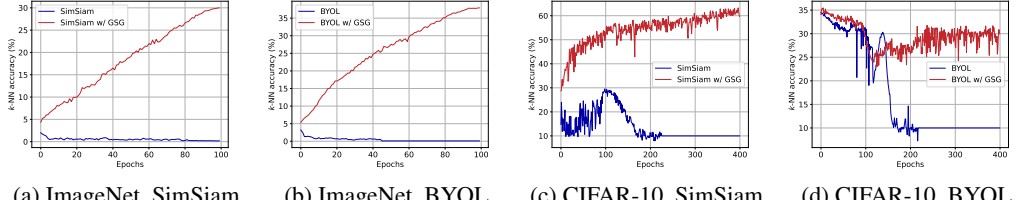

| (a) ImageNet, SimSiam | (b) ImageNet, BYOL | (c) CIFAR-10, SimSiam | (d) CIFAR-10, BYOL |

Figure 5: Preventing collapse. Unlike existing algorithms, algorithms to which GSG is applied do not collapse even when the predictor is removed.

Figure 4 shows the results of applying GSG, random stop-gradient, and reverse stop-gradient along with the existing algorithm for SimSiam and BYOL. We observe the $k$-NN accuracy at each epoch while training on ImageNet and CIFAR-10. First of all, in all cases, it can be seen that the algorithm applying our GSG outperforms algorithms using other methods. In addition, it can be seen that the performance of the existing algorithm and the algorithm to which random stop-gradient is applied are similar. When applying random stop-gradient, the number of loss terms is doubled, and half of them are randomly selected, so there is expected to be no significant difference from the existing algorithm. In the case of reverse stop-gradient, the performance drops significantly. This highlights the importance of using stop-gradient in a guided manner.

If we look at the case of CIFAR-10 (Figure 4c and Figure 4d), which has relatively much smaller data and is more challenging to train stably, we can obtain some more interesting results. First, when reverse stop-gradient is applied to SimSiam, it collapses, and the accuracy converges to $10\%$, which is the chance-level accuracy of CIFAR-10. However, this was not the case for BYOL. This implies that the momentum encoder can help prevent collapse. Note that SimSiam and BYOL are identical in our experimental setup, except that BYOL has a momentum encoder. In addition, in the case of the existing SimSiam and SimSiam with random stop-gradient, the fluctuation of the accuracy at the beginning of training is severe. However, it is relatively less in the case of SimSiam with GSG. This suggests that GSG can help the stability of training.

## 6.2 Preventing collapse

One of the expected effects of GSG is to prevent collapse by implicitly repelling each other away from negative pairs. It is known that SimSiam and BYOL collapse without a predictor [Chen and He, 2021, Grill et al., 2020]. Then, a natural question is whether SimSiam and BYOL will not collapse even if the predictor is removed when GSG is applied. Figure 5 reports the performance when the predictor is removed. In the case of CIFAR-10, we run up to 400 epochs to confirm complete collapse.

First, the existing algorithms collapse as expected, and the accuracy converges to the chance-level accuracy ($0.1\%$ for ImageNet and $10\%$ for CIFAR-10). Interestingly, however, our algorithms do not collapse. This shows that GSG contributes to training stability, which is our method's intrinsic advantage. Nevertheless, the final accuracy is higher when there is a predictor. This indicates that the predictor in SimSiam and BYOL contributes to performance improvement.

## 7 Conclusion

We have proposed implicit contrastive representation learning for visual SSL. In this methodology, while representations of positive pairs attract each other, representations of negative pairs are promoted to repel each other. It exploits the asymmetry of network architectures with source and target encoders without contrastive loss. We have instantiated the methodology and presented our guided stop-gradient method, which can be applied to existing SSL algorithms such as SimSiam and BYOL. We have shown that our algorithms consistently perform better than the benchmark asymmetric learning algorithms for various tasks and datasets. We have also shown that our algorithms are more robust to reducing the number of negative samples than the benchmark contrastive learning algorithms. Our empirical study has shown that our method contributes to training stability. We hope our work leads the community to better leverage the asymmetry between source and target encoders and enjoy the advantages of both contrastive learning and asymmetric learning.

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
