# OpenReview forum: "Implicit Contrastive Representation Learning with Guided Stop-gradient"
_NeurIPS.cc/2023/Conference — NeurIPS 2023 poster_

### Official Review · Reviewer_CW7J · 2023-06-27

**Soundness:** 3 good
**Presentation:** 3 good
**Contribution:** 3 good
**Rating:** 6
**Confidence:** 4

**Summary:**

The paper proposes the implicit contrastive learning algorithm, which uses the guided stop-gradient to push away negative samples without the uniformity term in the contrastive loss. By applying the method to the non-contrastive methods including SimSiam and BYOL, the paradigm combines the advantages of contrastive and non-contrastive algorithms and boosts the downstream performance in various downstream tasks.

**Strengths:**

1. The improvements brought by GSG are significant and the experiments are solid.
2. The main idea and insights of GSG are easy to follow.
3. The combination between the asymmetric architectures and contrastive loss is interesting.

**Weaknesses:**

1. Besides the guided stop-gradient, simply combing the asymmetric architectures and the contrastive loss (e.g., InfoNCE) seems would have the same effect. So what is the advantage of implicit contrastive loss? Is it possible that the advantages of GSG with small batch sizes are brought by the asymmetric architecture? More comparisons between the implicit contrastive loss and the explicit contrastive loss would make this paper more solid.
2. In Section 6.2, the authors show that GSG contributes to training stability by removing the predictors. However, the final accuracy is still far from the results with the predictor. Is it possible to show that the GSG stabilizes the training process of SimSiam with the predictor? It would be better to show more differences between SimSiam and SimSiam with GSG during the training process.
3. When selecting where to apply stop-gradient, the paper uses the distance in the projection layer. However, the loss is calculated in the prediction layer. Besides the brief explanations on page 4, it would be better to provide more theoretical or empirical evidence.

**Questions:**

See Weaknesses

---

> ### Author Rebuttal · Authors · 2023-08-09
>
> [W1] asymmetric architecture + contrastive loss
>
> MoCo is an algorithm that combines asymmetric architecture (stop-gradient, momentum encoder) and contrastive loss (InfoNCE). We refer the reviewer to Figure 2c in [1]. As shown in Table 4, even in the case of MoCo, the performance is not good when the batch size is small. The advantage of implicit contrastive loss is that, unlike contrastive loss, it works well with small batch sizes. From this example, we can see that this advantage comes from the implicit application of contrastive learning rather than from asymmetry.
>
> ---
>
> [W2] training stability
>
> As described in Lines 282-285 and Figure 4c, SimSiam w/ GSG shows a more stable learning curve in the training process. To quantify this, we measure how much the accuracy $x_t$ fluctuates at the beginning of the training. We take the first difference $y_t=x_t-x\_{t-1}$ and find the standard deviation of $\\{y_t\\}_{t=1}^{50}$. As can be seen in the following, SimSiam w/ GSG has the smaller standard deviation and is therefore more stable. Please also note that in the case of SimSiam w/ Reverse SG, it collapsed.
>
> |Algorithm|Std|
> |-|-|
> |SimSiam|11.306|
> |SimSiam w/ Random SG|9.755|
> |SimSiam w/ Guided SG|**5.527**|
>
> ---
>
> [W3]
>
> Since the predictor $h$ is a two-layer MLP head, it can be expressed as $h = f_2 \circ \phi \circ f_1$, where $f_1$ and $f_2$ are affine functions ($x \mapsto Ax+b$) and $\phi$ is the ReLU activation function ($x \mapsto \max(0, x)$). It is known that affine functions and ReLU are Lipschitz continuous. Note that a function $f: \mathbb{R}^n \rightarrow \mathbb{R}^m$ is Lipschitz continuous if there exists a constant $L$ such that $\lVert f(x) - f(y) \rVert_2 \leq L \lVert x- y \rVert_2$ for all $x, y \in \mathbb{R}^n$. Since the composition of Lipschitz continuous functions is also Lipschitz continuous, $h$ is Lipschitz continuous. So where $\lVert z - z' \rVert_2$ is small, $\lVert h(z) - h(z') \rVert_2$ is also small.
>
> ---
>
> [1] He, Kaiming et al., Momentum contrast for unsupervised visual representation learning, 2020, CVPR.

---

> > ### Comment · Reviewer_CW7J · 2023-08-20
> > **reply to the rebuttal**
> >
> > I thank the authors for the responses. I will maintain my original rating.

---

### Official Review · Reviewer_imWm · 2023-07-03

**Soundness:** 3 good
**Presentation:** 3 good
**Contribution:** 2 fair
**Rating:** 6
**Confidence:** 4

**Summary:**

This article presents a way to improve the learning of non-contrastive self-supervised learning methods such as BYOL and SimSiam.
It uses incorporates implicitly contrastive notions, by removing elements of the loss that may lead to close representations collapsing together. This modification is proven to improve the representations for BYOL and SimSiam with different experiments. In particular, variations of the algorithm with other Stop Gradients are shown to be detrimental to learning, and the utility of the algorithm to prevent collapse at low batch sizes or without projectors is shown.

**Strengths:**

The presented method is simple but provides consistent improvement for both BYOL and the SimSiam learning methods.

In particular, the method improves low batch sizes as well as when the predictor is removed.

The method is ablated to consider different Stop Gradients variations.

The article is clear and well written.

**Weaknesses:**

I find the article a bit lacking in ablations and inquiries about the method. For instance:
- I would have liked a variation of the algorithm with more than only 2 examples. Do using N examples for the implicit contrast improves further the method?
- And the method prevents collapse without a predictor, but the improvement is more unclear in the general case. Some measure of dimensionality for instance, or something similar, would have helped demonstrate the improvement of the method is due to helping a potential collapse of the representations. (I do not find the t-SNE visualisations to be convincing)

I am surprised by the accuracy results presented for the benchmark of the method. Exploring Simple Siamese Representation Learning, Chen et. al, 2021 provide in Table 4 accuracies for SimCLR and Moco which are much closer to the SimSiam accuracies than the one presented in this article. In particular, the End-to-End accuracies seem very low. For these reasons, I am a bit unconvinced by the benchmark.

I found the 3 Figures redundant for explaining the method. Similarly, I found the method to be a bit overexplained.

The paper in its present state is a bit light, which is why I am only considering a weak accept for now.



**Questions:**

Computing distances between representations is quite costly. I would expect a slowdown due to the computation for the method. Is it present, or is it compensated by the reduced number of computations needed by the Stop Gradients?

Have other alternatives, such as reweighting the "attracting" part of the loss rather than simply removing them been considered?

Do the authors have some hypotheses about the links between this work and explicit contrastive learning methods?

**Limitations:**

The authors have adequately addressed the limitations of the article, albeit in the appendix.

---

> ### Author Rebuttal · Authors · 2023-08-09
>
> [W1-1] using $N$ examples
>
> To use $N$ examples at once when constructing the loss, we need to create decision criteria like Equation (6) that considers $4{N \choose 2}$ distances together. We think it is hard to be done by a straightforward extension of our idea since when deciding which side to apply stop-gradient for each example, $N-1$ results from comparison with other examples may not be consistent.
>
> In our idea, we drive representations away from each other by pairing each example with another. So it is natural to consider two examples at a time when constructing the loss. Since we calculate the cost by aggregating the losses over the batch, we after all use as many examples as the batch size in each gradient update.
>
> ---
>
> [W1-2] measure of dimensionality
>
> Please see [G1] in the global response above.
>
> ---
>
> [W2] accuracy for the benchmark
>
> As for the performance of the benchmark algorithms in Table 4, we used those reported in SimCLR and MoCo's original papers (please refer to Table B.1 in [1] and Figure 3 in [2]). The SimCLR and MoCo accuracies in Table 4 in [3] you mentioned are obtained under different settings than ours.
>
> For SimCLR, the higher accuracy is that of a much larger batch size 4096. This again shows that the performance is greatly affected by the batch size in the case of explicit contrastive learning.
>
> For MoCo, the performance the table is reporting is for MoCo v2 [4], which is an improved version of MoCo. Also, in the MoCo framework, negative samples are from a dictionary decoupled from the batch, but the dictionary size is not written in the paper.
>
> ---
>
> [Q1] distance computation cost
>
> There is additional computation due to computing distance. However, it is a vectorized computation that is performed once for each iteration. Passing through a ResNet is a more dominant cost.
>
> ---
>
> [Q2] reweighting the loss terms
>
> The experiment in Section 6.1 can be seen as the reweighting. When we construct the loss, we need to choose one from $L_1=\\{\frac{1}{2}\mathcal{D}(p_{11}, \text{sg}(z_{12})), \frac{1}{2}\mathcal{D}(p_{12}, \text{sg}(z_{11}))\\}$ and one from $L_2=\\{\frac{1}{2}\mathcal{D}(p_{21}, \text{sg}(z_{22})), \frac{1}{2}\mathcal{D}(p_{22}, \text{sg}(z_{21}))\\}$ (refer to Figure 2a).
>
> This can be relaxed to put two-point distribution over each set. For each set, if we give a probability of 1 to the term chosen with the idea of GSG and 0 to the other term, it becomes GSG. Conversely, if probabilities of 0 and 1 are given, it becomes Reverse SG, and if probabilities of 0.5 and 0.5 are given, it becomes Random SG.
>
> ---
>
> [Q3] link between implicit and explicit contrastive learning
>
> Please see [G2] in the global response above.
>
> ---
>
> [1] Chen, Ting et al., A simple framework for contrastive learning of visual representations, 2020, ICML.
>
> [2] He, Kaiming et al., Momentum contrast for unsupervised visual representation learning, 2020, CVPR.
>
> [3] Chen, Xinlei and He, Kaiming, Exploring simple siamese representation learning, 2021, CVPR.
>
> [4] Chen, Xinlei et al., Improved baselines with momentum contrastive learning, 2020, arXiv preprint arXiv:2003.04297.

---

> > ### Comment · Reviewer_imWm · 2023-08-11
> >
> > I thank the authors for their answers to my questions, notably on the benchmark and the computation cost.
> >
> > - For my question about using $N$ examples, I meant a generic value of examples and not only 2. I apologize for the use of $N$ as I did not mean the batch size in particular. Even using 3 examples could allow for a more complex formulation, by increasing the number of distances considered and thus the strength of the implicit contrast.
> >
> > - The addition of the relative variance does show an improvement of dimensionality.
> >
> > Some of my concerns have been answered, however I still find the article light since they are no ablations or extensions of the method. This method is still effective at low batch sizes which makes it a worthwile intermediate between contrastive and non-contrastive learning, so I am keeping my rating as a weak accept.

---

> > > ### Author Response · Authors · 2023-08-11
> > > **Response to Reviewer imWm**
> > >
> > > We thank the reviewer for the response. We hope the following addresses the reviewer's concerns.
> > >
> > > ---
> > >
> > > Our ablation study can be found in Section 6. There we provide:
> > >
> > > 1. To find out the effect of **Guided SG**, it was compared with other variants, Random SG and Reverse SG.
> > >
> > > 2. To find out how much the **predictor** contributes, it was compared with the case where the predictor was removed.
> > >
> > > 3. By comparing SimSiam and BYOL, the influence of the **momentum encoder** (only in BYOL) could be investigated.
> > >
> > > Note that the components of SimSiam w/ GSG are two encoders, one predictor, and the GSG method. In the case of BYOL w/ GSG, a momentum encoder is one of the two encoders. Therefore, we have dealt with the major components in our ablation study.
> > >
> > > ---
> > >
> > > The extension the reviewer mentioned requires bringing in other ideas to deal with new situations.
> > >
> > > For instance, if there are 3 examples, there are 2 reference examples $x_2$ and $x_3$ for one example $x_1$. When determining which of the two projections $z_{11}$ and $z_{12}$ to apply stop-gradient, the result considering the relationship with $x_2$ and the result considering the relationship with $x_3$ may be different. That is, one may be $z_{11}$, and the other may be $z_{12}$. At this time, we need another criterion to break the tie. More situations must be considered if we consider more than 3 examples simultaneously.
> > >
> > > However, introducing these additional ideas is not a straightforward extension. We believe different papers require different levels of extension. Since this paper aims to propose a simple but transformative idea, trying to add more to the current algorithm can make it overly complex and is beyond the scope of the paper.

---

### Official Review · Reviewer_Wcuk · 2023-07-04

**Soundness:** 3 good
**Presentation:** 3 good
**Contribution:** 3 good
**Rating:** 6
**Confidence:** 4

**Summary:**

This paper proposes a novel SSL technique that can be applied on top of SimSiam or BYOL to select where to apply asymmetric predictor. This method first computes embeddings (before the predictor) and compute relevant distances. Then based on this distance, it chooses where to apply the predictor. Experiment results show improvement of this method over SimSiam/BYOL baseline.


**Strengths:**

1. The idea is well motivated, based on the dynamics of SimSiam/BYOL representation space.
2. Experiment shows the effectiveness of the proposed method. I appreciate full evaluation including linear probe, kNN, transfer learning, and detection.
3. Ablation of several different strategies for choosing predictors is very insightful.


**Weaknesses:**

1. Overall the innovation is limited. It is an interesting trick over SimSiam/BYOL, but not transferrable to general SSL architecture.
2. Experiment is questionable. Using batch size 512 is not considered optimal with ImageNet pre-training. In fact, the imagenet linear probe numbers reported for the BYOL baseline are lower than the original paper, with this batch size (it is supposed to be only -0.5% lower than using 4k batch size). I understand that it is only trained on 8 NVIDIA A100 GPUs. But it is possible that hyperparameters are no longer optimal.


**Questions:**

1. Does this method effectively double the batch size? How exactly each `pairs` of images are sampled? What is memory/compute overhead?

---

> ### Author Rebuttal · Authors · 2023-08-09
>
> [W1] generalizability
>
> Asymmetry is an important topic in recent self-supervised representation learning [1]. Algorithms that utilize asymmetry such as SimSiam, BYOL, SwAV, and DINO are continuously emerging. In this paper, we showed that asymmetry, which was previously introduced to prevent collapse, can help improve performance if exploited more actively. Since this is the first paper in this direction, we first applied our ideas to the relatively simple SimSiam and BYOL. However, we believe that the idea of implicitly performing contrastive learning using asymmetry can be applied to other algorithms in a different way or provide a clue to the emergence of new algorithms.
>
> ---
>
> [W2] performance in the original paper of BYOL
>
> We guess the performance of BYOL in the original paper you mentioned is from Table 1 in [2]. The performance in the table is the result of a different setting including longer training (please refer to Section 3.3 in [2]). Our paper focuses on the relative performance gap between the algorithms. So the set of hyperparameters of the algorithms are all set to that of the original SimSiam paper for apple-to-apple comparison.
>
> ---
>
> [Q1] sampling pairs of images
>
> We put the given batch and a shuffled batch side by side and pair the images one by one (please refer to Appendix A). The batch size remains the same because negative samples are paired within a given batch.
>
> ---
>
> [1] Wang, Xiao et al., On the importance of asymmetry for siamese representation learning, 2022, CVPR.
>
> [2] Grill, Jean-Bastien et al., Bootstrap your own latent-a new approach to self-supervised learning, 2020, NeurIPS.

---

> > ### Comment · Reviewer_Wcuk · 2023-08-18
> >
> > I thank the authors for the response.
> > All my questions are addressed. I will update my score accordingly.

---

### Official Review · Reviewer_cNZ6 · 2023-07-05

**Soundness:** 3 good
**Presentation:** 3 good
**Contribution:** 3 good
**Rating:** 7
**Confidence:** 4

**Summary:**

The paper introduces the Guided Stop-Gradient (GSG) method that can be applied to SSL algorithms that adopt asymmetric dual encoders such as BYOL and SimSiam in order to boost their performance and stabilize their training. The idea of the GSG is to augment the loss function to attract different views of two different images instead of just one as it is done in BYOL and SimSiam. However, the method does not explicitly repel the representations of the views of the different images but carefully selects which positive views to attract with a stop-gradient operation. Experiments with pretraining are performed on BYOL and SImSiam models using ImageNet and CIFAR10 datasets and their trained model is applied on several downstream tasks.


**Strengths:**

- The method is simple and practical to implement.
- Results are consistent and demonstrate increased performance on pretraining and downstream tasks.
- Using this method, models can be trained with lower batch size which is a plus.


**Weaknesses:**

I believe that the paper in the current form lacks analysis (and intuition) that would help understanding why the method works better than the baselines. For example, I would like to see more analysis on the structure of the learned representation space, and the effect of the choice of the batch size and number of “negative” images added to the loss. Given that the experiments are performed on smaller datasets, this is feasible to add. See also my questions below.


**Questions:**

I have the following questions:

1. I find that choices of k in Section 5.1. unintuitive. In the case of CIFAR10, wouldn’t a large k give more insights into how well the latent space is separated? On the other hand, using k=200 for ImageNet seems very large. Could you report the variation in labels for the retrieved neighbors, i.e., is the majority vote significant? This would be interesting to analyze also in case of CIFAR10 for larger k.

2. In linear probing, you train the classification layer for the same amount of epochs as the backbone and on a much larger batch size, which sounds a lot. Is there a reason for increasing the batch size to 4096? Have you tried with just a few iterations and smaller batch size?

3. I was wondering if you could add some discussion and analysis that would help better understand why GSG works well. It would be particularly interesting to understand why GSG isn’t affected by changes in the batch size. My hypothesis is that this is because the model isn’t explicitly penalized for misplacing (potentially many) negative examples. This is partially addressed in Appendix E, however, I find tSNE visualization subjective. Do you have any numbers supporting the separation of clusters, for example, std of representations of each class? Another experiment could be to add more images to your loss (x3, x4, …), would that hurt or boost the performance?

4. I am wondering what is the dimension of the representation space? Just to be clear, in my understanding you use z in all your experiments?

Minor:
5. In Section 5.2. it is not clear whether you finetune the backbone or use the frozen model.

6. When applying random stop-gradient in Section 6.1, do you randomly choose two terms out of 4 or do you randomly choose one term for each of the images? i.e, randomly choosing either first or second term and either third or fourth in Equation 2?

7. I believe that the claim in line 284 that the fluctuation of the accuracy in the beginning of the training of SimSiam is less severe when the model is trained with GSG is too strong without supporting numbers. Could you add some variance analysis?

8. Section 3 is helpful for understanding the idea of the model but it would be great if you could make it explicit that x1 and x2 are different images.


**Limitations:**

Code is provided but the paper checklist is not. Societal impacts nor limitations are not discussed.

---

> ### Author Rebuttal · Authors · 2023-08-09
>
> [W1] analysis on the learned representation space
>
> Please see [G1] in the global response above.
>
> ---
>
> [Q1] choices of $k$ for $k$-NN
>
> We tried to make the experimental settings identical to previous studies for easy and fair comparison. So for the value $k$, we used the default value in [1] for the CIFAR-10 experiments and the value used in [2, 3] for the ImageNet experiments. It is hard to report the label variation for the retrieved neighbors since it varies per example. Table 11 of [3] also shows the results for $k=20$ in ImageNet experiments. So we also report the results for $k = 20$ below. Performance was still better when GSG was applied.
>
> |Algorithm|$k$-NN acc. (%)|
> |-|-|
> |SimSiam|55.6|
> |SimSiam w/ GSG|**62.8**|
> |BYOL|60.8|
> |BYOL w/ GSG|**66.2**|
>
> ---
>
> [Q2] number of iterations and batch size for linear evaluation
>
> For linear evaluation, we used the same number of iterations and batch size as in SimSiam's paper (please refer to Section A in [4]). In linear evaluation, even after a few iterations, the accuracy comes close to the final accuracy. For example, in the case of SimSiam, the accuracy after 90 epochs is about 67.9% and the accuracy after 10 epochs is about 64.3%. We also tried batch size 256, and it gave a slightly lower accuracy (~1%).
>
> ---
>
> [Q3] analysis and discussion on why GSG works well
>
> Please see [G1] and [G2] in the global response above. Regarding adding more images to the loss, it is hard to scale naturally, given the nature of our method of comparing distances between projections. If there are N examples, we need to compare $4{N \choose 2}$ distances. Also, when considering one example, the results obtained by comparing with the remaining N-1 negative examples may not be consistent.
>
> ---
>
> [Q4] dimension of the representation space
>
> In many algorithms (SimSiam, BYOL, SimCLR, etc.) including ours, the encoder is a backbone plus a projector, which is a shallow MLP head. In actual evaluation, we use the representations (from the backbone), which are closely related to the projections (from the projector). Attaching a projector after the backbone is a common practice to improve performance. We refer the reviewer to Line 108 and Footnote 2. So the dimension of the representation space is same to the input dimension of the projector. For CIFAR-10, it is 512, and for ImageNet, it is 2048 (please refer to Lines 42 and 49 in Appendix).
>
> ---
>
> [Q5] transfer learning evaluation mode
>
> We used the frozen model (please refer to Line 232).
>
> ---
>
> [Q6] random stop-gradient
>
> We randomly chose one of the four equations in Equation (6).
>
> ---
>
> [Q7] supporting numbers for the fluctuation of the accuracy
>
> To measure the fluctuating degree of the accuracy $x_t$ at the beginning of training, we take the first difference $y_t=x_t-x\_{t-1}$ and find the standard deviation of $\\{y_t\\}_{t=1}^{50}$. The following shows that SimSiam w/ GSG has a much smaller standard deviation. Thus, its accuracy fluctuates less than other algorithms.
>
> |Algorithm|Std|
> |-|-|
> |SimSiam|11.306|
> |SimSiam w/ Random SG|9.755|
> |SimSiam w/ Guided SG|**5.527**|
>
> ---
>
> [Q8] $x_1$ and $x_2$ are different images.
>
> Thank you for your suggestion. We will add this in the revised version.
>
> ---
>
> [L1] paper checklist, societal impacts, limitations
>
> In this NeurIPS, the paper checklist was not attached to the back of the paper, but was to be filled in on the OpenReview system. You can see it at the top of this page. Societal impacts and limitations are discussed in Section G.
>
> ---
>
> [1] Susmelj, Igor et al., Lightly, 2020.
>
> [2] Wu, Zhirong et al., Unsupervised feature learning via non-parametric instance discrimination, 2018, CVPR.
>
> [3] Caron, Mathilde et al., Deep clustering for unsupervised learning of visual features, 2018, ECCV.
>
> [4] Chen, Xinlei and He, Kaiming, Exploring simple siamese representation learning, 2021, CVPR.

---

> > ### Comment · Reviewer_cNZ6 · 2023-08-14
> > **reply to the rebuttal**
> >
> > I thank the authors for the thorough rebuttal. I especially appreciate the discussion and results in [G1] and [G2], and the extra results to answer my questions. Based on these comments I recommend to accept the paper. I increased my rating accordingly.

---

### Author Rebuttal · Authors · 2023-08-09

Dear reviewers,

We thank you for your careful reading and constructive feedback. Your comments will help us improve the quality of the paper. We have detailed responses to each reviewer individually below. We also write responses to some common questions here. We write [Gx], [Wx], [Qx], and [Lx] for a global response, weakness, question, and limitation reference, respectively. If additional explanations are needed, we will be happy to provide them.

---

[G1] analysis of why our method works well

In addition to qualitative analysis, t-SNE in Section E, we performed the following quantitative analysis to support this. We first defined between-class, within-class, and relative variance and investigated whether our method would improve representation quality in terms of the relative variance.

Let $\mathcal{X}$ be the set of all representations, and $N = \vert \mathcal{X} \vert$. For each label $i$ ($1 \leq i \leq K$), let $\mathcal{C_i}$ be the set of all representations with label $i$, and $N_i = \vert \mathcal{C_i} \vert$. Then, the total mean $\mathcal{X}$ and the class mean $\mathcal{C_i}$ are written as

$\bar{x} = \frac{1}{N} \sum_{x \in \mathcal{X}} x, \quad \bar{x_i} = \frac{1}{N_i} \sum_{x \in \mathcal{C_i}} x.$

We define the between-class variance $v_b$ and the within-class variance $v_w$ as follows.

$v_b = \frac{1}{K} \sum_{i \in [K]} d(\bar{x_i}, \bar{x}), \quad v_w = \frac{1}{K} \sum_{i \in [K]} \left( \frac{1}{N_i} \sum_{x \in \mathcal{C_i}} d(x, \bar{x}_i) \right)$,

where $[K]=\\{1,2,\cdots,K\\}$, and $d(\cdot, \cdot)$ is the Euclidean distance. So $v_b$ is the average distance between a class mean and the total mean, and $v_w$ is the average distance between a representation and its class mean. Then, the relative variance $v_r$ is written as
$v_r = v_b / v_w.$

The following is the relative variance obtained for the algorithms.

|Algorithm|Relative Variance|
|-|-|
|SimSiam|2,323|
|SimSiam w/ GSG|**2.821**|
|BYOL|2.458|
|BYOL w/ GSG|**3.049**|

The result shows our method increases between-class variance relative to within-class variance. Note also that this is similar to the goal of linear discriminant analysis (LDA) [1].

---

[G2] discussion on why our method works well

Self-supervised learning (SSL) can basically be seen as harnessing two forces. One is attracting force between positive pairs, and the other is repelling force between negative pairs (if any). Where these two forces are in balance, representations are formed.
- If the attracting force is too large, all representations come together (collapse).
- If the repelling force is too large, it repels even the positive pairs (sampling bias from not knowing the labels).

First, contrastive learning gained momentum in SSL research, but it had drawbacks such as performance degradation due to sampling bias and the need for a large batch size to secure many negative samples.

Later, non-contrastive learning using only positive pairs, such as SimSiam or BYOL, gained more and more attention, but these algorithms raise the question of whether there might be room for improvement by using a contrastive effect.

Our implicit contrastive learning uses negative sampling like contrastive learning, but on the surface, it uses only the attracting force like non-contrastive learning. So it can be seen as an attempt to find a sweet spot between these two domains. To this end, we leverage asymmetry, which many non-contrastive learning algorithms introduce to prevent collapse initially.

As shown in Section 5.3, our algorithms performed more robustly than the contrastive learning algorithms at small batch sizes. In many explicit contrastive learning algorithms, N-1 negative examples are explicitly repelled from one example. However, in our method, one negative example is paired with one example in an implicit way. This may be why batch size reduction does not affect our algorithms much.

Also, as shown in Sections 5.1-2, it performed better than the non-contrastive learning algorithms. This may be because our method contributes to better separation of clusters, as shown in t-SNE and [G1]. As such, we confirmed the untapped potential of the use of asymmetry through this study.

---

[1] Härdle et al., Applied multivariate statistical analysis, 2019, Springer Nature.

---

### Decision · Program_Chairs · 2023-09-21

**Decision:**

Accept (poster)

**Comment:**

Post rebuttal, reviewers have achieved a consensus to accept the work. The AC checked all the materials and concurs that the paper has done a reasonable exploration in the direction of non-contrastive self-supervised representation learning, by implicitly doing contrastive learning with guided stop-gradient. The proposed method has been validated by a series of experimental results, including the additional ones during the rebuttal period. Please incorporate necessary changes in the final version.